# Interventions that improve maternity care for immigrant women in the UK: protocol for a narrative synthesis systematic review

Gina Marie Awoko Higginbottom,[1] Catrin Evans,[1] Myfanwy Morgan,[2] Kuldip Kaur Bharj,[3] Jeanette Eldridge,[4] Basharat Hussain[1]

► Prepublication history and additional material are available. To view these files please visit the journal online (http://dx.doi.org/10.1136/bmjopen-2017-016988).

[1]School of Health Sciences, University of Nottingham, Nottingham, UK
[2]Primary Care and Public Health Sciences, King's College London, London, UK
[3]Faculty of Medicine and Health, University of Leeds, Leeds, UK
[4]Research and Learning Services, University of Nottingham, Nottingham, UK

**Correspondence to**
Dr Basharat Hussain; basharat.hussain@nottingham.ac.uk

## ABSTRACT

**Introduction** A quarter of all births in the UK are to mothers born outside the UK. There is also evidence that immigrant women have higher maternal and infant death rates and of inequalities in the provision and uptake of maternity services/birth centres. The topic is of great significance to the National Health Service because of directives that address inequalities and the changing patterns of migration to the UK. Our main question for the systematic review is 'what interventions exist that are specifically focused on improving maternity care for immigrant women in the UK?' The primary objective of this synthesis is to generate new interpretations of research evidence. Second, the synthesis will provide substantive base to guide developments and implementation of maternity services/birth centres which are acceptable and effective for immigrant women in the UK.

**Methods and analysis** We are using a narrative synthesis (NS) approach to identify, assess scientific quality and rigour, and synthesise empirical data focused on access and interventions that enhance quality of maternity care/birth centres for the UK immigrant women. The inclusion criteria include: publication date 1990 to present, English language, empirical research and findings are focused on women who live in the UK, participants of the study are immigrant women, is related to maternity care/birth centres access or interventions or experiences of maternity. In order to ensure the robustness of the NS, the methodological quality of key evidence will be appraised using the Center for Evidence-Based Management tools and review confidence with CERQual (Confidence in the Evidence from Reviews of Qualitative Research). Two reviewers will independently screen studies and extract relevant evidence. We will synthesise evidence studying relationships between included studies using a range of tools.

**Dissemination** Dissemination plan includes: an e-workshop for policymakers, collaborative practitioner workshops, YouTube video and APP, scientific papers and conference presentations.

## Strengths and limitations of this study

► This systematic review uses a narrative synthesis (NS) approach to search, assess scientific quality and rigour, and synthesise findings from empirical research focused on access and interventions to enhance quality of maternity care/birth centres for women of immigrant background.

► A NS (1) facilitates understanding and acknowledges the broader impact of theory and context-related variables including ethnicity, social and economic position, and geography; (2) enables insights into how differences are determined in the reported outcomes as the consequence of diversity in research design and immigrant women in reproductive phase; (3) provides findings that are helpful in the development and implementation of maternity care/birth centres.

► This systematic review will review quantitative, qualitative as well as mixed-methods research data.

► This review will use the 'four dimensions of access' theoretical framework in the healthcare context.

► This is not a meta-synthesis thus limiting our access to the raw data.

by a dynamic interplay of variables among an increased number of new, small and scattered, multiple-origin, transnationally connected, socio-economically differentiated and legally stratified immigrants'[5] (p 1024) who started arriving in the UK after 1990. It has been argued that current level of superdiversity in the UK has resulted in a huge challenge for understanding and meeting individualised healthcare needs.[5]

Offering relevant and effective maternity healthcare to immigrant women in the UK is crucial for achieving maximum health and well-being potentials.[2 3 6–8] Absence of culturally relevant and safe healthcare can result in negative outcomes ranging from inadequate communication to life-risking events with serious consequences in maternity.[9]

## INTRODUCTION

The UK is in a period of superdiversity[1] with more diverse populations accessing maternity care.[1–4] Superdiversity 'is distinguished

> **Box 1  Gulliford's theory of access in healthcare.**
>
> ► Service availability
> ► Utilisation of services and barriers to access, which includes personal, financial and organisational barriers
> ► Relevance, effectiveness and access
> ► Equity and access.
>
> These four dimensions are extremely pertinent in respect of providing a theoretical lens in order to explore the experience of immigrant women. We use this[12] theory because primacy is given to the notion of equity of access as the most significant dimension. The National Health Service seeks to provide a universal and egalitarian service, consequently this might be the most profound and significant dimension of access. Our review provides the opportunity for a nuanced and comprehensive exploration of the four facets of access.

For example, immigrant women are over-represented in mortality statistics particularly in maternal and peri-natal mortality.[10 11] While recent reviews have focused on specific maternity care aspects[6–8] they have ignored a comprehensive conceptualisation of access as opposed to Gulliford *et al*,[12] and failed to embrace current superdiversity.[1] Reconfiguration and design of maternity care/birth centres in the UK requires integration of all these aspects.

Our review uses Gulliford's theory of access[12] (box 1).

Gulliford's theoretical framework on access interfaces with National Institute for Health and Clinical Excellence (NICE)[13] research recommendations on access and models of service provision. NICE regards some immigrant women as having complex social factors. It notes that women who are pregnant and are recent immigrants, refugees, asylum seekers or who find it hard to read and speak English may not be fully benefitting from antenatal healthcare services. The reason behind this may include lack of knowledge about the health services and/or poor communication with staff delivering healthcare. The NICE guidance suggests that '*healthcare professionals should help support these women's uptake of antenatal care services by, using a varied of means to communicate with women, telling women about antenatal care services and how to use them, and undertaking training in the specific needs of women in these groups*' (p 16). The guidance further recommends that healthcare staff be given specific training in meeting the needs of the group.

Our aim is to undertake a narrative synthesis (NS) of a wide range of empirical literature including synthesis of non-peer-reviewed literature to offer stakeholders an understanding regarding access and maternity services/birth centre interventions (National Health Service (NHS) and non-NHS) directed at immigrant women in the UK.

Our objectives for this synthesis include:
► To search, assess scientific quality and synthesise quantitative, qualitative and mixed-methods empirical papers on the subject.
► To search, appraise and synthesise grey literature as well as non-empirical reports.

► To recognise further users of knowledge and methods for knowledge transfer.
► To disseminate the synthesis findings via strategic end-of-grant knowledge transfer.

Our ultimate aim is to establish the present knowledge base and produce significant suggestions for policy and practice in future, thus helping to achieve equity in healthcare.

To reach our aim, we are using a project advisory group (PAG), established during the finalisation of the review questions and preliminary planning for sharing results. The PAG would continue to help during the whole project cycle. The group will be active partner in the research process and engage in all steps in the review.

## METHODS AND ANALYSIS

We will follow a NS approach to systematic review. According to Popay *et al*, NS is 'an approach to the systematic review and synthesis of findings from multiple studies that relies primarily on the use of words and text to summarise and explain the findings of the synthesis.'[14] The focus of a NS approach is to synthesise review findings interpretatively instead of undertaking meta-analysis of the evidence. NS will allow the review to embrace inter-disciplinary and methodological wide-ranging research to map access and interventions in maternity care/birth centres. The main findings from this synthesis will then be used to explain why and how maternity care/birth centre interventions focused on immigrant women have been implemented and how these interventions address access and inequalities experienced by immigrant women. Our preliminary scoping review (devised by a research librarian) has helped identify a range of quantitative and qualitative studies focused on access and interventions which enabled us to determine that a synthesis does not yet exist with a specific focus on our review question.

NS consists of four components: (A) theory development to explain how, why and for whom the intervention works; (B) producing an initial synthesis of the results from the included studies; (C) finding the links in the data; (D) evaluation of the rigour of the NS. These components are not exclusive and the NS uses an iterative approach. In each component, various tools and techniques can be used, which of course depend on the nature of the research data.

## ELIGIBILITY CRITERIA
### Study characteristics

This review will review qualitative, quantitative and mixed-methods research evidence. Therefore, no study will be excluded based on its type. All research evidence published in English language from 1990 to present will be eligible for inclusion in the review. We are focusing on 1990 as the immigration patterns in the UK changed from 1990 onwards. Prior to 1990, immigration to the UK was predominately from India, Pakistan, Bangladesh and

Caribbean. The review will include peer-reviewed and grey literature.

## POPULATION

We will review empirical and grey literatures that report on maternity care/birth centres for immigrant women in the UK. In this review, we define an immigrant woman—one who is born outside the UK, has foreign citizenship and has moved to the UK to stay temporarily (at least a year) or has intention to settle for the long term.[15] Therefore, economic migrants (both skilled and unskilled), refugees, asylum seekers, students and illegal immigrants' population will be included.[16] Literature on British-born Black, Asian and Minority Ethnic participants would be ineligible for inclusion in the review.

## INTERVENTION

By intervention we mean 'a combination of program elements or strategies designed to produce behaviour changes or improve health status among individuals or an entire population.'[17] The interventions that we plan to review are those that immigrants participate in during the antenatal, intrapartum or postnatal period in healthcare or social or community setting. Therefore, only interventions relevant to pregnancy and the postpartum period (up to 12 months postdelivery) will be reviewed. Interventions must be specifically focused on immigrant women and those aimed at the general population will be excluded.

## CONTEXT

Maternity health services/birth centre-related interventions that are offered in the UK. Interventions focused on immigrants in any setting, that is, hospital and community will be included.

## OUTCOMES

The primary outcome from this project would be a NS of literature relating to maternity care/birth centres for immigrant women in the UK which would be published in high-impact journal, for example, *Social Science and Medicine, Journal of Health Services Research and Policy*, as well as open-access journals (eg, *BMC Pregnancy & Childbirth, BMJ Open Access*). Key findings and recommendations would be shared with relevant stakeholders through online workshop, seminars and meetings. Regarding secondary outcome, this project would offer a crucial base to guide development and implementation of maternity care/birth centre services which are acceptable and effective for immigrant women in the UK.

### Data sources and search strategy

The following given literature databases will be explored for relevant articles published between 1 January 1990 and 30 June 2017; the searching and retrieval of results will be completed between February and June 2017. The

quoted date range for each database and preferred host system are recorded as follows:
► Medline (1946 to present, Ovid)
► Embase (1980 to present, Ovid)
► PsycINFO (1806 to present, Ovid)
► HMIC (1979 to latest monthly update, Ovid)
► MIDIRS (1971 to latest monthly update, Ovid)
► CINAHL (to present, EBSCOHost)
► ASSIA (1976 to present, ProQuest)
► POPline (http://www.popline.org)
► Web of Science (1900 to present)
► Scopus (1970 to present)

An experienced research librarian will construct the detailed search strategies for each database and conduct the searches after review by the entire team. The Ovid MEDLINE search strategy will be tailored for each database to apply related controlled concepts, MeSH terms, keywords and search techniques (see online supplementary table 1). We will also review the reference lists of included studies for related citations and additional hand-searches will be undertaken in major relevant journals (eg, *J Immigr Health*) and as published by topical research groups. BH and JE will implement the search strategy independently (independent double screening), and meet to review any discrepancies or dissonance. Ambiguous papers will be reviewed by the entire team.

The search for grey literature will include thesis repositories (eg, e-Theses, ProQuest Dissertations), internet searches, Google Scholar, citation searches, clinical trial registers, research funders' listings of research projects and reviews of reference lists. Grey literature will be screened for relevance and categorised using the National Information Centre on Health Services Research system.[18]

## DATA MANAGEMENT

The review will use Endnote Bibliographic management software to manage the searches. All search hits will be directly imported into the Endnote and duplicates will be removed by the two reviewers independently.

## SELECTION PROCESS

We will use the PRISMA (Preferred Reporting Items for Systematic Reviews and Meta-Analyses) flow chart to (figure 1) document the review steps.[19] We will employ a three-step process: (1) screening of the studies; (2) initial categorisation of the studies; (3) getting full-text files, finalising studies and final categorisation, employing independent double screening. The screening of peer-reviewed articles includes screening of titles, and screening of abstracts with the attached screening tool (see online supplementary table 2). The screening tool will be piloted by two reviewers independently.

Grey literature documents will be screened for their relatedness, and categorised using the categories of grey literature established by the National Information Centre on Health Services Research and Health Care Technology

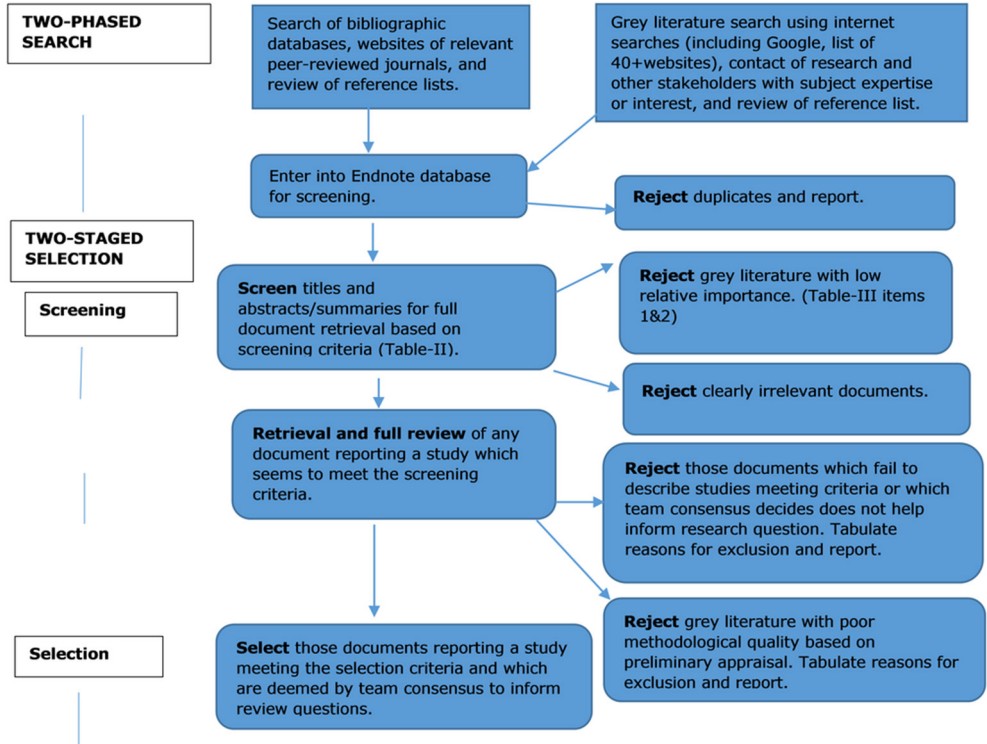

**Figure 1** PRISMA (Preferred Reporting Items for Systematic Reviews and Meta-Analyses) flow chart showing review steps.

at the National Library of Medicine (see online supplementary table 3).[18]

The two reviewers will independently extract evidence from all included studies, and any dissonance will be addressed through discussion, if required, involving the third reviewer to reach consensus. Data will be sought on variables such as which migrant group(s) the intervention(s) is focused, length of migration, type of intervention, setting of intervention (hospital/community), duration of intervention and intervention making agency (NHS, social care, voluntary group).

Four researchers experienced in numerous forms of systematic reviewer having undertaken formal training and conducted subsequent funded reviews[20–22] will be involved in data extraction (GMAH, CE, KKB, BH). A relevancy appraisal will be undertaken by first reviewing the title and abstract (GMAH, CE, KKB, BH) and then retrieving potentially relevant articles for further assessment to meet our inclusion criteria and with entire team deciding final inclusion. The scientific quality of key literature will be assessed using the Center for Evidence-Based Management (CEBMa) tools (eg, critical appraisal of a survey, critical appraisal of a qualitative study, critical appraisal of a case study)[23] and for assessment of confidence in the NS review findings with CERQual (Confidence in the Evidence from Reviews of Qualitative Research).[24] We are using critical appraisal tools of CEBMa as it has a full range of tools as opposed to Critical Appraisal Skills Programme (CASP), for example, CASP does not have a tool for survey research.

## MISSING DATA

Attempts will be made to contact study authors via email to obtain clarification if data are incomplete in the study report and we will allow a delay of 6weeks to receive a response following two email attempts.

## Quality appraisal

We plan to assess quality and the robustness of the synthesis through the following ways:

► Weight of evidence: Popay et al[14] identify the weight of evidence approach as a tool that can be used to assess the scientific quality and rigour of the synthesis results.

► Reflecting critically: Popay et al[14] suggested that a narrative summary of the synthesis should be given and includes the following: (1) synthesis methodology—this should have a special focus on the limitations and their potential influence on the findings; (2) data used: for quantitative papers—its quality, validity, reliability and generalisability. For qualitative papers—indicate possible sources of bias. This systematic review will use Lincoln and Guba's principles of confirmability, credibility, dependability and transferability.[25]

► Any assumptions made

► What disagreements and uncertainties were identified, and how these were addressed

► Identification of the fields where currently there is weak or absence of scientific evidence. Additionally, further areas of research should be indicated.

► Consideration and discussion on the 'thick' and 'thin' evidence,[20] including commentary on any similarity and/or difference in the evidence (GMAH, MM, CE, KKB, BH). 'Thick' evidence offers adequate clarity to achieve an understanding of the topic of our interest, whereas 'thin' evidence is unable to provide appropriate clarity to improve our knowledge of the phenomenon of interest detail to an understanding of the phenomenon.[14]

## Strategy for data synthesis

We have planned a narrative (descriptive) synthesis for this review for all types of studies. Patterns derived from the narrative description and comparison of the literatures will allow us to recognise the elements that influence maternity interventions and delivery of maternity care services/birth centres. These elements will be synthesised into major themes pertaining to enablers and hindrances that design interventions related to immigrant women and maternity care/birth centre services. We will use conceptual and thematic analysis using a range of clustering and networking tools. In addition to tabulation, we will use analytical software for further grouping (ie, atlas.ti7 by ATLAS.ti GmbH, Berlin, Germany).

## DISSEMINATION

A multilevel knowledge translation (KT) plan will be used so that key messages are strategically delivered to key stakeholders. We will facilitate an online workshop with key decision makers/stakeholders to ensure policy-relevant messages and use widely accessible technology (social networking, webinars) to maximise impact.

The systematic review team comprises individuals who have good engagement in hospital-based/community healthcare services for immigrant women. KT will continue via meetings with the women and community groups and workshops for immigrant/refugee health.

Theoretical and practical contribution will happen via publication of results in high-impact international journals, such as *J Immigr Minor Health*, *Sociology of Health and Illness*, *Social Science and Medicine*, *Journal of Health Services Research and Policy* and open-access journals (eg, *BMC Pregnancy & Childbirth*, *BMJ Open Access*).

**Contributors** GMAH conceptualised the review and drafted the original grant proposal with CE, MM and KKB. JE developed and implemented the scoping review strategy. BH drafted the manuscript for publication with input from other team members. All authors approved the final version of the protocol for publication.

**Funding** This systematic review is funded by the UK National Institute for Health Research (NIHR) Health Services and Delivery Research Programme (Grant No. HS&DR-15/55/03). Along with funding, NIHR also contributed in peer reviewing the funding proposal containing the review protocol. Any amendments (if required) in the protocol would be done in consultation with NIHR. Sponsor: Department of Health (DoH)-NIHR (543808).

**Competing interests** None declared.

**Provenance and peer review** Not commissioned; externally peer reviewed.

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
