## [Reviewer comments · BMJ Open]

ARTICLE DETAILS

TITLE (PROVISIONAL)	Interventions that improve maternity care for immigrant women in the United Kingdom (UK): protocol for a narrative synthesis systematic review
AUTHORS	Higginbottom, Gina; Evans, Catrin; Morgan, Myfanwy; Bharj, Kuldip; Eldridge, Jeanette; Hussain, Basharat

VERSION 1 - REVIEW

REVIEWER	Rebecca Garcia University of Bedfordshire, England.
REVIEW RETURNED	20-Apr-2017

GENERAL COMMENTS	Thank you for giving me the opportunity to review this timely protocol on maternity interventions for immigrant women in the UK. This narrative review at present states it will focus on 'immigrant' women - i.e. those from overseas, who settle in the UK. This will substantially limit the scope and outcomes of this review, as some British-born ethnic minority groups suffer from adverse birth outcomes and inequalities too. It would be more useful to clarify explicitly which groups of women are included, for example, British-born BAME, illegal immigrants, or foreign-born BAME. Although the date range from 1990 is justified to represent the changes in migration patterns to the UK, there have also been policy changes in the last decade, so addressing inequalities and equity of access is a more recent focused occurrence. I think it would be useful to the reader acknowledge the changing emphasis toward addressing health inequalities - this will help contextualise the findings in the forthcoming results paper.
--

REVIEWER	Reem Malouf Oxford University
REVIEW RETURNED	09-May-2017

GENERAL COMMENTS	Thank you for asking me to review this interesting protocol. I have a few suggestions: Please remove citations from the abstract. I suggest to summarize the review inclusion/exclusion criteria and remove indication to table 1 from the abstract. I suggest the addition of the primary and secondary objectives of the
--

	review to the abstract. Context: I suggest the addition of birth centers. For outcomes: I suggest to report your outcomes as primary and secondary outcomes. Medline search: Lines 23 to 50 were all performed to limit the search by date, is it possible to perform the same limitation by using one line such as "limit the search to 1990"?
--	---

VERSION 1 – AUTHOR RESPONSE

BMJ review comments	Response from authors
Reviewer(s)' Comments to Author: Reviewer:1 This narrative review at present states it will focus on 'immigrant' women - i.e. those from overseas, who settle in the UK. This will substantially limit the scope and outcomes of this review, as some British-born ethnic minority groups suffer from adverse birth outcomes and inequalities too. It would be more useful to clarify explicitly which groups of women are included, for example, British-born BAME, illegal immigrants, or foreign-born BAME.	We acknowledge your observation that there are maternity care inequalities in the British-born minority ethnic groups. However, NIHR has funded us to undertake narrative synthesis on 'immigrant' women. Please see link below for NIHR website and PROSPERO registration. http://www.dc.nihr.ac.uk/themed-reviews/better-beginnings.htm (study 68, p. 52) http://www.crd.york.ac.uk/PROSPERO/display_record.asp?ID=CRD42015023605 We have clarified in the article that our focus is on foreign born women including illegal immigrant. British- born BAME women are excluded. Please see p. 5
Although the date range from 1990 is justified to represent the changes in migration patterns to the UK, there have also been policy changes in the last decade, so addressing inequalities and equity of access is a more recent focused occurrence. I think it would be useful to the reader acknowledge the changing emphasis toward addressing health inequalities - this will help contextualise the findings in the forthcoming results paper.	There is renewed interest in addressing health inequalities. Please see for example NHS document on 'National Maternity Review: BETTER BIRTHS, Improving outcomes of maternity services in England, A Five Year Forward View for maternity care'. https://www.england.nhs.uk/wp-

	content/uploads/2016/02/national-maternity-review-report.pdf
Reviewer:2 Please remove citations from the abstract.	We have removed citations from the abstract. Please see p. 1
I suggest to summarize the review inclusion/exclusion criteria and remove indication to table 1 from the abstract.	We have summarised the review inclusion/exclusion criteria and removed indication to table 1 from the abstract. Please see p. 1
I suggest the addition of the primary and secondary objectives of the review to the abstract.	We have added primary and secondary objectives in the review abstract. Please see p. 1
Context: I suggest the addition of birth centers.	We have taken your comment on board and added birth centres in 'context'. e.g please see. p. 5
For outcomes: I suggest to report your outcomes as primary and secondary outcomes.	We have reported outcomes as primary and secondary outcomes. Please see. p. 5
Medline search: Lines 23 to 50 were all performed to limit the search by date, is it possible to perform the same limitation by using one line such as "limit the search to 1990"?	Lines 23-52 are not relevant for the overall strategy and should be disregarded. They were used to enable download of the answer set in manageable tranches of records (file download size limitations applied by the Ovid host) for import to EndNote, and are not relevant to the strategy per se.